# Breakthrough SARS-CoV-2 Infections after Vaccination in North Carolina

**DOI:** 10.3390/vaccines10111922

**Published:** 2022-11-13

**Authors:** Diane Uschner, Matthew Bott, William H. Lagarde, Joseph Keating, Hazel Tapp, Andrea A. Berry, Austin L. Seals, Iqra Munawar, John Schieffelin, Joshua Yukich, Michele Santacatterina, Mihili Gunaratne, Lida M. Fette, Brian Burke, Greg Strylewicz, Sharon L. Edelstein, Amina Ahmed, Kristen Miller, John W. Sanders, David Herrington, William S. Weintraub, Michael S. Runyon

**Affiliations:** 1The Biostatistics Center, George Washington University, Rockville, MD 20852, USA; 2Division of Pediatric Endocrinology, Department of Pediatrics, WakeMed Health and Hospitals, Raleigh, NC 27610, USA; 3Department of Tropical Medicine, School of Public Health and Tropical Medicine, Tulane University, New Orleans, LA 70118, USA; 4Department of Family Medicine, Atrium Health Carolinas Medical Center, Charlotte, NC 28262, USA; 5Department of Pediatrics, Center for Vaccine Development and Global Health, University of Maryland School of Medicine, Baltimore, MD 21201, USA; 6Division of Cardiology, Department of Internal Medicine, Wake Forest School of Medicine, Winston-Salem, NC 27109, USA; 7Division of Infectious Diseases, Department of Internal Medicine, Wake Forest School of Medicine, Winston-Salem, NC 27109, USA; 8Section of Infectious Disease, Department of Medicine, Tulane University School of Medicine, New Orleans, LA 70112, USA; 9Department of Pediatrics, Atrium Health Levine Children’s Hospital, Charlotte, NC 28203, USA; 10MedStar Health Research Institute, Georgetown University, Washington, DC 20007, USA; 11Department of Emergency Medicine, Atrium Health Carolinas Medical Center, Charlotte, NC 28262, USA

**Keywords:** SARS-CoV-2 vaccination, cumulative incidence, booster vaccination, age, rural county, Omicron, Delta, BNT162b2, Ad26.COV2.S, mRNA-1273

## Abstract

We characterize the overall incidence and risk factors for breakthrough infection among fully vaccinated participants in the North Carolina COVID-19 Community Research Partnership cohort. Among 15,808 eligible participants, 638 reported a positive SARS-CoV-2 test after vaccination. Factors associated with a lower risk of breakthrough in the time-to-event analysis included older age, prior SARS-CovV-2 infection, higher rates of face mask use, and receipt of a booster vaccination. Higher rates of breakthrough were reported by participants vaccinated with BNT162b2 or Ad26.COV2.S compared to mRNA-1273, in suburban or rural counties compared to urban counties, and during circulation of the Delta and Omicron variants.

## 1. Introduction

While COVID-19 vaccines prevent or attenuate the severity of SARS-CoV-2 infection, little is known about risk factors for breakthrough infection [1,2]. This remains a critical gap in knowledge as breakthrough infections have a significant impact on essential workforce capacity in settings with high vaccine uptake, including healthcare, and serve as a secondary reservoir for transmission to others at risk. We characterize the overall incidence and risk factors for breakthrough infection among fully vaccinated participants in a large COVID-19 surveillance cohort—the North Carolina (NC) COVID-19 Community Research Partnership (CCRP) [3].

## 2. Materials and Methods

The NC CCRP is an observational cohort study assessing COVID-19 symptoms, test results, vaccination status, and risk behavior via daily email or text surveys. Adults 18 years and older were enrolled between 13 April 2020 and 14 August 2021 at six NC healthcare systems (http://www.covid19communitystudy.org/, 30 September 2022). The participating study sites invited participants from inside and outside their healthcare systems via email and patient portal messaging, study websites, and during community outreach events. This study was approved by the Wake Forest School of Medicine Institutional Review Board. The analysis includes data through 3 January 2022.

### 2.1. Data Collection

Self-reported data comprised a short questionnaire at enrollment, daily surveys (link sent each day via e-mail or text message), and periodic short supplementary surveys. Demographic data, including age, race/ethnicity, healthcare worker status, and COVID-19 infection prior to enrollment, and address were collected at enrollment. Counties of residence were classified as urban, suburban, or rural based on average population density per square mile as calculated by the NC Rural Center based on the 2020 U.S. Census [4]. Potential COVID-19 symptoms that were queried in the surveys included fever, chills, cough, shortness of breath, fatigue, muscle pain, headache, loss of taste/smell, sore throat, congestion/runny nose, nausea/vomiting, and diarrhea. At enrollment, participants consented to share their electronic health records (EHR). Records were only accessed if a participant was also a patient in one of the participating study sites. Sites used their identity management service to confirm that a consented participant was a patient in their health care system. Breakthrough infection was defined according to the Centers for Disease Control and Prevention guidance as “... the detection of SARS-CoV-2 RNA or antigen in a respiratory specimen collected from a person ≥14 days after receipt of all recommended doses of an FDA-authorized COVID-19 vaccine [5]”.

### 2.2. Vaccination

We defined SARS-CoV-2 vaccination as participant self-report of receiving two doses of either the Pfizer BioNTech BNT162b2 or Moderna mRNA-1273 vaccine, or one dose of the Janssen non-replicating viral vector vaccine Ad26.COV2.S. Vaccination status was confirmed for the subset of participants with accessible EHR data [6]. A participant was considered fully vaccinated fourteen days after they had received one dose of Janssen (J&J) or two doses of an mRNA vaccine (Pfizer BioNTech BNT162b2 or Moderna mRNA-1273). A participant was considered to have received a booster if they received a vaccine dose at least 60 days after their second dose of mRNA (or missing or unknown) vaccine or after their first dose of J&J. The vaccination brand was defined as the brand of the first vaccine dose received. Participants reporting vaccination with another vaccine brand (e.g., Astra Zeneca or Other) were excluded. The first participant considered fully vaccinated entered the analysis on 15 January 2021, and the last participant on 30 October 2021.

### 2.3. Primary Outcome: Self-Reported Testing for SARS-CoV-2 Infection

In the current analysis, the primary outcome was weeks until the first self-reported infection (positive SARS-CoV-2 antigen or nucleic acid amplification test) occurring ≥14 days after full vaccination. Participants were able to self-report test results from any COVID-19 viral test they took as part of their work, social life, or after developing symptoms, including tests that were conducted at a testing center. Time to breakthrough was defined as time from full vaccination to first self-reported infection. Participants were censored if they withdrew from the study, at the date of their last status update, or at the end of the analysis period on 3 January 2022.

### 2.4. Time-Varying Covariates

Time-varying covariates were incorporated on a participant by calendar week basis. The vaccination proportion in the county of residence by calendar week was estimated as the cumulative number of “full” vaccination doses up to that calendar week [7], divided by the total adult population in the county of residence. Vaccination proportion was shifted by three calendar weeks, anticipating that antibodies are fully formed two weeks after full vaccination, and allowing an additional week for exposures to take place and self-reported infections to arise. Percent mask usage was defined as the proportion of days in a calendar week the participant reports the use of a face mask if meeting others outside the household within six feet distance, or reporting no contacts. Booster shot was coded as a binary variable that was true starting the week the participant received the booster vaccine. The rationale for shifting the variable by one week was to allow time for an exposure to take place and self-reported infections to arise, similar to the vaccination proportion. The time frame for the Delta period was defined as calendar weeks 26 to 46 (28 June 2021 to 21 November 2021). The time frame for the Omicron period was defined as calendar weeks 49 to 53 (5 December 2021 to 3 January 2022) [8].

### 2.5. Missing Data

Nearly 80% of the cohort reported at least once per calendar week for all the weeks of their follow-up. Two participants with missing sex and participants with vaccine brand other than Janssen, Pfizer, and Moderna were excluded from the analysis. Participants who did not specify their race/ethnicity at enrollment were classified as “Non-Hispanic Other”. Participants who did not indicate that they consider themselves healthcare workers were considered non-healthcare workers. Participants who did not complete a daily survey in a calendar week were considered to have used a face mask < 90% of the time.

### 2.6. Statistical Analysis

Cox proportional hazards models with fixed and time-varying covariates were used to estimate hazard rates. Univariate and multivariate models were run, with all variables assessed in the univariate models included in the multivariate model. Event rates were calculated as the total number of events divided by one hundredth the total duration of follow-up of all participants in the cohort. Time to event was modeled using Cox proportional hazards models using robust variance estimates to account for data clustered by participant [9]. Only main effects were entered into the model. Wald 95% confidence intervals (CIs) for hazard rates and Wald-test *p*-values were calculated. *p*-values lower than 5% were considered significant. All analyses used R version 4.0.3 [10].

## 3. Results

Study population characteristics are summarized in Table 1. Of 15,808 eligible participants, 638 (4.0%) reported a positive SARS-CoV-2 test after vaccination, reflecting an event rate of 6.7 events per 100 person-years of follow-up. The total SARS-CoV-2 positivity rate for the same time period statewide for North Carolina was 10.5 cases per 100 residents; this same population had a 53.5% vaccination rate [7]. Infections were symptomatic in 593 (93%) cases.

Median and interquartile follow-up times following full vaccination were 29.3 interquartile range (IQR = 21.7–42.7) weeks among infected participants and 31.6 (IQR = 21.7–36.6) weeks among uninfected participants. The cumulative incidence of breakthrough infection was 6.95% over 45 weeks of follow-up following full vaccination (Figure 1). While the time period prior to the Delta variant made up 42.9% of total person-years in follow-up, only 3.7% of events were observed in that period. The period when the Delta variant was the most prevalent in NC made up 49.3% of the total person-years in follow-up, and accounted for 60.7% of events. The period when Omicron was the most prevalent variant reflected only 7.7% of total person-years, but accounted for 35.5% of events.

In the multivariable analysis (Table 1), age 45 and older was associated with lower risk of breakthrough infection. hazard ratio (HR) and 95% CI of age 45–64 vs. 18–44 was 0.70 (0.59–0.82); age 65+ vs. 18-44 was 0.41 (0.32–0.54). Prior SARS-CoV-2 infection, HR (95% CI) = 0.58 (0.39–0.85); higher rates of face mask use, HR (95% CI) = 0.66 (0.56–0.79); and receipt of a booster vaccination, HR (95% CI) = 0.33 (0.27–0.40) were also associated with a lower risk of breakthrough infections. Compared to those vaccinated with Moderna mRNA1273, participants vaccinated with Pfizer/BNT BNT162b2 or J&J Ad26.COV2.S had a higher risk of breakthrough infection; the HR (95% CI) of BNT162b2 vs. mRNA1273 was 1.35 (1.10–1.66); the HR (95% CI) of Ad26.COV2.S vs. mRNA1273 was 1.74 (1.24–2.44). Participants from rural and suburban counties had a higher risk of breakthrough infections; HR (95% CI) of suburban vs. urban was 1.33 (1.08- 1.64); and rural vs. urban was 1.24 (1.01–1.53). Participants had a higher risk of infection during circulation of the Delta and Omicron variants compared to earlier time periods: HR (95% CI) for Delta vs. Pre-Delta was 3.54 (2.34–5.35), and for Omicron vs. Pre-Delta 16.68 (10.05–27.68). There was no association of breakthrough infection with sex, race/ethnicity, healthcare worker status, or vaccination rate in the county of residence.

When the multivariable analysis was stratified by age (Appendix A), breakthrough infections were lowest in those who received Moderna vaccines, and boosters lowered the risk of breakthrough at similar rates across all three age groups. The youngest age group had a higher risk of breakthrough infection during the Omicron period, HR (95% CI) = 27.42 (12.58–59.76), which is in line with their higher overall risk of breakthrough infection in the un-stratified model, HR (95% CI) = 16.63 (10.02–27.59).

## 4. Discussion

Among vaccinated individuals, the magnitude of risk for breakthrough infection and possible strategies to mitigate that risk are of great importance. The rate of breakthrough infections increased over time, consistent with increasing community spread in concert with waning vaccine effectiveness. Our data support previous reports of higher effectiveness of Moderna mRNA-1273 relative to Pfizer/BNT BNT162b2 [11,12], and even more so relative to J&J Ad26.COV2.S, and re-iterate the dramatically higher risk for breakthrough infections during the Omicron surge [12]. The risk associated with the Omicron period likely reflects several factors, including the infectivity of the Omicron variant, a higher community prevalence of infection, and waning vaccine immunity [13]. The significantly lower rates of breakthrough infection associated with mask wearing and receipt of a booster are consistent with previous reports and highlight specific measures that may minimize the risk for COVID-19 despite prior vaccination [14,15]. Similarly, higher rates among younger individuals may reflect more frequent or higher risk exposures, including those related to childcare or to differences in occupational or social exposures. This reinforces the importance of avoiding settings where respiratory viral transmission may more easily occur and using risk mitigation measures when exposures cannot be avoided. A point of ongoing debate relates to the possible additional protection provided by naturally acquired immunity in combination with vaccination [16]. Our data reveal a 42% reduction in the relative hazard for breakthrough among vaccinated participants who self-reported prior SARS-CoV-2 infection. A particularly interesting and unexpected finding was the increased risk of breakthrough infection among participants residing in rural and suburban counties compared with those from urban areas. We initially hypothesized this might be due to differences in vaccination rates by county, but the differences persisted even after adjusting for county vaccination rates.

The daily survey responses capture whether a test was conducted and the result of the test, but does not capture the detail of where, how, or why the test was performed. Naturally, this type of self-directed testing that does not follow a pre-described testing schedule is more likely to detect symptomatic infections and under-detect asymptomatic infections. Moreover, participants may take tests and fail to report them in the system, which may happen more often with negative tests. In weeks when participants did not have any status update, including no self-reported infection, infections were treated as absent, likely resulting in an underestimation of the incidence of SARS-CoV-2 infection.

In any observational study, there are significant limitations related to bias or residual confounding, generalizability, and power such that results should be interpreted with caution. In addition, our cohort includes a large proportion of healthcare workers and there may be important differences among this group compared with the general population with regard to study interest and engagement, exposure profiles, and risk mitigation behaviors. This degree of oversampling of healthcare workers was not intentional, but rather the likely result of the fact that enrollment was based at large healthcare systems. Furthermore, the difference in vaccination release dates was not accounted for in the models. Nevertheless, many of our findings corroborate the work of others, supporting the validity of the data.

## 5. Conclusions

In summary, this real-world analysis adds to the understanding of risk factors associated with breakthrough SARS-CoV-2 infections and highlights opportunities for mitigation. Specifically, the lower rates of breakthrough infection associated with face mask use and receipt of a booster highlight specific measures that individuals can take to minimize their risk for COVID-19.

## Figures and Tables

**Figure 1 vaccines-10-01922-f001:**
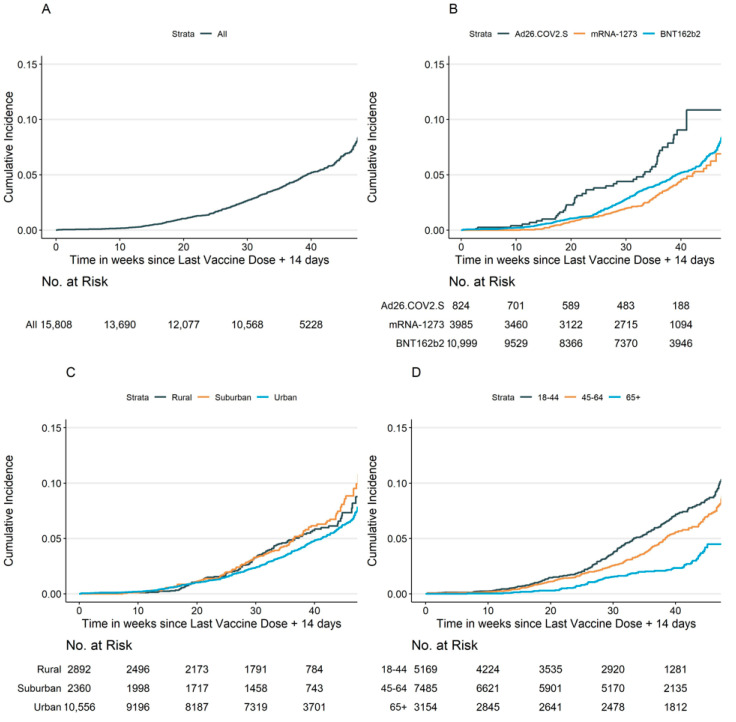
Cumulative incidence curves of breakthrough infection among participants who reported full vaccination (1 minus the unadjusted Kaplan–Meier risk). Below each graph is the number at risk at each time point for the first self-reported symptomatic positive SARS-CoV-2 test, starting from full vaccination: (**A**) Overall; (**B**) By vaccine product; (**C**) By county classification; (**D**) By age stratum.

**Table 1 vaccines-10-01922-t001:** Infection rate per 100 person-years overall and by factors, and factors associated with SARS-CoV-2 infection after vaccination.

Characteristics				Univariate		Multivariate	
Fixed Covariates	N (%)	Of Events (%)	Event Rate (95% CI)	Hazard Ratio (95% CI)	*p*-Value	Hazard Ratio (95% CI)	*p*-Value
Overall	15808 (100%)	638 (4%)	6.7 (6.2–7.2)				
Age							
18–44 (Reference group)	5169 (33%)	244 (4.7%)	8.6 (7.6–9.8)				
45-64	7485 (47%)	315 (4.2%)	6.9 (6.2–7.7)	0.71 (0.60–0.84)	<0.0001	0.70 (0.59–0.82)	<0.0001
65+	3154 (20%)	79 (2.5%)	3.7 (3.0–4.6)	0.35 (0.27–0.45)	<0.0001	0.41 (0.32–0.54)	<0.0001
Sex							
Female (Reference group)	11631 (74%)	471 (4%)	6.7 (6.1–7.3)				
Male	4177 (26%)	167 (4%)	6.6 (5.7–7.7)	0.97 (0.82–1.15)	0.7387	1.11 (0.93–1.32)	0.2399
Race/Ethnicity							
Non-Hispanic White (Reference group)	14854 (94%)	611 (4.1%)	6.8 (6.2–7.3)				
Non-Hispanic Black	400 (3%)	8 (2%)	3.8 (1.9–7.6)	0.72 (0.36–1.43)	0.3484	0.7 (0.35–1.41)	0.3163
Hispanic	165 (1%)	6 (3.6%)	7.0 (3.1–15.6)	1.24 (0.56–2.75)	0.6041	0.96 (0.43–2.16)	0.9289
Non-Hispanic Other	389 (2%)	13 (3.3%)	6.2 (3.6–10.7)	1.04 (0.61–1.78)	0.8785	1.01 (0.59–1.73)	0.9789
Healthcare Worker Status							
No (Reference group)	10518 (67%)	374 (3.6%)	6.0 (5.4–6.6)				
Yes	5290 (33%)	264 (5%)	8.1 (7.1–9.1)	1.14 (0.96–1.35)	0.1268	1.12 (0.92–1.36)	0.2647
Vaccination Product							
Moderna-mRNA-1273 (Reference group)	3985 (25%)	120 (3%)	5.1 (4.2–6.1)				
Pfizer/BNT-BNT162b2	10999 (70%)	473 (4.3%)	7.0 (6.4–7.7)	1.24 (1.02–1.51)	0.0351	1.35 (1.10–1.66)	0.0042
J&J-Ad26.COV2.S	824 (5%)	45 (5.5%)	10.2 (7.6–13.6)	2.22 (1.59–3.10)	<0.0001	1.74 (1.24–2.44)	0.0014
Prior COVID Infection							
No (Reference group)	14743 (93%)	610 (4.1%)	6.8 (6.3–7.4)				
Yes	1065 (7%)	28 (2.6%)	4.8 (3.3–6.9)	0.79 (0.54–1.15)	0.2129	0.58 (0.39–0.85)	0.0052
County Classification							
Urban (Reference group)	10554 (67%)	410 (3.9%)	6.3 (5.7–7.0)				
Suburban	2361 (15%)	111 (4.7%)	8.1 (6.7–9.7)	1.32 (1.07–1.62)	0.0087	1.33 (1.08- 1.64)	0.0076
Rural	2893 (18%)	117 (4%)	7.0 (5.8–8.4)	1.20 (0.98–1.47)	0.0756	1.24 (1.01–1.53)	0.0396
Time-varying covariates							
Vaccination Rate in County of Residence 3 weeks prior							
≤60% (Reference group)							
>60%				0.81 (0.61–1.07)	0.1425	0.85 (0.64–1.13)	0.2692
Mask Usage in week prior							
≤90% (Reference group)							
>90%				0.61 (0.52–0.73)	<0.0001	0.66 (0.56–0.79)	<0.0001
Delta Time Frame *							
Not in Delta Time Frame (Reference group)							
In Delta Time Frame				0.87 (0.71–1.07)	0.1814	3.54 (2.34–5.35)	<0.0001
Omicron Time Frame **							
Not in Omicron Time Frame (Reference group)							
In Omicron Time Frame				4.19 (3.33–5.28)	<0.0001	16.68 (10.05–27.68)	<0.0001
Receipt of booster shot							
No (Reference group)							
Yes				0.35 (0.29–0.43)	<0.0001	0.33 (0.27–0.40)	<0.0001

* Delta Time Frame set as between 28 June 2021 and 21 November 2021. ** Omicron Time Frame set as between 5 December 2021 and 3 January 2022, the end of the observation period for this analysis.

## Data Availability

The datasets used and/or analyzed during the current study are available from the corresponding author on reasonable request. Results of the COVID-19 CCRP are being disseminated on the study website (https://www.covid19communitystudy.org, 30 September 2022) as well as in publications and presentations in medical journals and at scientific meetings. At end of the study, the databases will be made publicly available in a de-identified manner according to CDC and applicable U.S. Federal policies.

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
