# Peer review of "Breakthrough SARS-CoV-2 Infections after Vaccination in North Carolina"

_vaccines, 2022, doi:10.3390/vaccines10111922_

Round 1

Reviewer 1 Report

The purpose of the manuscript was to evaluate the factors associated with breakthrough infection following vaccination and boosting with the BNT162b2, As26.COV2.S, or mRNA-1273 approved vaccines. This study included a large sample size with over 15,000 participants with 4% (over 600) reporting a positive SARS-CoV-2 test. The authors evaluate factors such as age, ethnicity, vaccine manufacturer, prior infection, and county classification. Following multivariant analysis, the authors determined that young age (18-44), vaccination with the Pfizer or J&J vaccine, no prior infection, living in suburban or rural counties, reduced mask wearing, and infection during Delta or Omicron surges resulted in a higher risk of SARS-CoV-2 infection. The results from this study align with previous reports. Overall, the study uses a large data set to analyze factors that influence breakthrough infection in vaccinated individuals in a specific region. The number of participants was large but there was a low number of cases in certain categories, which reduced the power of the study. Compared to other studies, the authors look broadly at breakthrough infections rather than focusing on a group with a particular ailment. Overall, the study lacks novelty, but it adds to the growing knowledge of critical areas of focus or targeted demographic for boosters to reduce vaccine breakthrough, which needs to be reported.

Major comments:

11.   It would be useful if the authors performed additional multivariant analysis but broke down the data based on age. Currently the analysis evaluated each covariant as an individual entity. It would be interesting to analyze the significant factors (vaccination product, prior infection, county, mask usage, etc) within an age group. This may provide insight into different behaviors of the different age groups which could be related to why they are more or less likely to have a breakthrough infection.

22.       For factors with more than two groups, was the reference group varied to determine if there were difference between all the groups? Example: Pfizer vs J&J, 45-64 vs 65+, etc.

Minor comments:

11.       Figure 2 should be Figure 1. Please adjust.  

22.       In the Figure 2 legend, please include a description of what the numbers represent and if those numbers relate to the weeks listed on the graph.

33.       More detail and clarity regarding the categorization parameters should be provided to the audience with regard to how County Classifications into Urban, Suburban, and Rural groups was made. Population density estimates and generalized determination should be indicated within the text within the data collection section.

44.       Within the discussion section, cite and briefly summarize other studies mentioned in lines 185-186 that support the data reported within this report.

Author Response

Reviewer 1

  • It would be useful if the authors performed additional multivariant analysis but broke down the data based on age. Currently the analysis evaluated each covariant as an individual entity. It would be interesting to analyze the significant factors (vaccination product, prior infection, county, mask usage, etc) within an age group. This may provide insight into different behaviors of the different age groups which could be related to why they are more or less likely to have a breakthrough infection.
    Response: To address your request to stratify the multivariate analysis by age, we have now added Supplementary Table 1. This table demonstrates that protective effects observed with vaccination and receipt of booster were held across age groups. The most noteworthy outcome of these models is that the youngest age group had a higher risk of breakthrough infection during the Omicron period. This is now reported in the Results (Lines 153-159).
  • For factors with more than two groups, was the reference group varied to determine if there were difference between all the groups? Example: Pfizer vs J&J, 45-64 vs 65+, etc.
    Response: Each category in our models were only compared to the reference category. The confidence intervals in Table 1 can be used to help the reader determine differences for each group compared to each other.
  • Minor comments: In the Figure 2 legend, please include a description of what the numbers represent and if those numbers relate to the weeks listed on the graph. More detail and clarity regarding the categorization parameters should be provided to the audience with regard to how County Classifications into Urban, Suburban, and Rural groups was made. Population density estimates and generalized determination should be indicated within the text within the data collection section. Within the discussion section, cite and briefly summarize other studies mentioned in lines 185-186 that support the data reported within this report.
    Response: We have addressed all of your minor comments by adding more clarity to the Figure 2 description (line 166), adding more detail of how counties were classified (line 53), and referencing other supporting studies in lines 180-182.0

Reviewer 2 Report

I only have two minor suggestions:

- What was included in univariate and multivariate analysis?

- Time periods are specific to vaccines since they had different launch dates (especially JNJ relative others) Age better be treated continuous variables: can be clarified in limitations or accounted for in methods (preferably)

Author Response

Reviewer 2:

  • What was included in univariate and multivariate analysis?
    Response: We have added a sentence clarifying what is included in the multivariate models to the Methods (lines 113-114).

  • Time periods are specific to vaccines since they had different launch dates (especially JNJ relative others)
    Response: This is now addressed in the Conclusion as a limitation (lines 216-217)
  • Age better be treated continuous variables: can be clarified in limitations or accounted for in methods (preferably)
    Response: We prefer to stratify age on the typical cutpoints used in COVID-19 research, in particular so the reader can easily see differences in the youngest and oldest study participants.

Reviewer 3 Report

Diana Uschner and colleagues describe the results of long-term studies investigating SARS-CoV-2 breakthrough infections in North Carolina. Impressive in scale, a cohort of 15808 ensures sound results, presented studies provide an interesting insight into effectiveness of vaccination in prevention of COVID-19 infection. Overall, I like the manuscript, it is brief but informative, well written and the results are clearly presented. To make the paper even more sound, there are number of issues that could addressed.

Major points:

1/ The studies focused on fully vaccinated cohort. It is a valid approach, however, some reference to entire population would be necessary to put the data into the context. For instance, the key information of the paper is that 638 out of 15808 fully vaccinated individuals become infected during the study period (that is roughly 4%). What was the infection rate for the same time period for entire population of North Carolina)? Please provide some estimate based on the publicly available data (ideally overall, vaccinated and non-vaccinated.

2/ The composition of the cohort is biased towards healthcare professionals. They make up 33 per cent of the cohort, and probably even more in the under-65 age groups. Was this intentional, or are healthcare professionals more willing to participate in such studies? Please provide some explanation in the Discussion.

3/ Concerning the Discussion, please discuss following issues:

a/ Why individuals living in rural and suburban areas were at higher risk of infection? Sheer population density of urban areas would suggest opposite results, could you please put this observation into the local context?

b/ Discuss your work in the context of others. There are many similar studies looking at similar issues in different regions and countries. How does it compare?

4/ Please provide statistic in the Kaplan-Meier charts.

Minor points:

Please clarify the term “breakthrough infection” in the introduction. I do realize that a reader of the journal “Vaccines" may be expected to be familiar with such a term, but it is key to understanding the paper, and in times of pandemic, more lay readers may read your article.

Author Response

Reviewer 3:

  • The studies focused on fully vaccinated cohort. It is a valid approach, however, some reference to entire population would be necessary to put the data into the context. For instance, the key information of the paper is that 638 out of 15808 fully vaccinated individuals become infected during the study period (that is roughly 4%). What was the infection rate for the same time period for entire population of North Carolina)? Please provide some estimate based on the publicly available data (ideally overall, vaccinated and non-vaccinated.
    Response: We have now pulled data on the number of positive COVID cases in North Carolina (NC) during our time period (01/15/2021 - 01/03/2022) from the NC Department of Health and Human Services (NCDHHS) website. This allowed us to estimate the rate per 100 residents using the NC population in 2021 (10.55 million) from the United States Census Bureau. The COVID positivity rate in NC was 10.5 per 100 residents which is slightly higher than the event rate of 6.7 per 100 person years we observe in our data. We were not able to stratify the state data by vaccination status but we were able to calculate the percentage of fully vaccinated residents using NCDHHS data, 53.5% of NC residents were vaccinated during out study time frame compared with 100% in our study cohort, which likely explains our lower rate. This was added to the Results (lines 124-126).
  • The composition of the cohort is biased towards healthcare professionals. They make up 33 per cent of the cohort, and probably even more in the under-65 age groups. Was this intentional, or are healthcare professionals more willing to participate in such studies? Please provide some explanation in the Discussion.
    Response: We have added an explanation in the Discussion (lines 210-216).
  • Concerning the Discussion, please discuss following issues: a/ Why individuals living in rural and suburban areas were at higher risk of infection? Sheer population density of urban areas would suggest opposite results, could you please put this observation into the local context? b/ Discuss your work in the context of others. There are many similar studies looking at similar issues in different regions and countries. How does it compare?
    Response: We have added to the Discussion our initial hypothesis regarding the rural and suburban effect on breakthrough infection and how we adjusted our analysis to include vaccination rates by county but still observed the effect (lines 190-194). We have also added additional references to the Discussion.
  • Please provide statistic in the Kaplan-Meier charts.
    Response: We created Kaplan-Meier curves with 95% confidence intervals but they were cluttered and were harder to asses visually than those without the intervals displayed. Hazard ratios with 95% confidence intervals are available in Table 1 so we removed the confidence intervals from the Kaplan-Meier curves.
  • Please clarify the term “breakthrough infection” in the introduction. I do realize that a reader of the journal “Vaccines" may be expected to be familiar with such a term, but it is key to understanding the paper, and in times of pandemic, more lay readers may read your article.
    Response: We have added a description and reference to the Methods clarifying what our definition of “breakthrough infection” (lines 60-63).
  •